# Identifying Factors for Selecting Land over Maritime in Inter-Regional Cross-Border Transport

**Shinya Hanaoka** [1,*] **, Takuma Matsuda** [2] **, Wataru Saito** [1] **, Tomoya Kawasaki** [3] **and Takashi Hiraide** [1]

1   School of Environment and Society, Tokyo Institute of Technology, 2-12-1 O-okayama, Meguro-ku, Tokyo 152-8550, Japan; saito.w.ab@gmail.com (W.S.); hiraide.t.ac@m.titech.ac.jp (T.H.)
2   Faculty of Commerce, Takushoku University, 3-4-14 Kohinata, Bunkyo-ku, Tokyo 112-8585, Japan; tmatsuda@ner.takushoku-u.ac.jp
3   School of Engineering, The University of Tokyo, 7-3-1 Hongo, Bunkyo-ku, Tokyo 113-8656, Japan; kawasaki@sys.t.u-tokyo.ac.jp
*   Correspondence: hanaoka@ide.titech.ac.jp; Tel.: +81-3-5734-3468

**Abstract:** Several cross-border land corridor projects have been implemented worldwide, because land transport is a vital alternative to international maritime transport in inter-regional transport. Maritime transport generally costs less than land transport, but it is much slower. Nonetheless, land transport can be more appropriate than maritime under certain situations. This study aims to identify factors that can help select between these two modes in long-distance inter-regional cross-border transport; to this end, a Tobit model is employed to estimate the dependent variable, i.e., the land ratio of origin–destination pairs between countries and/or areas. Eight variables are identified as significant: distance, export of manufacturing commodity, landlocked country/area, neighboring country/area, country risk, infrastructure level, port-access time, and maritime transport frequency. We also find that geographical conditions, country relationship, and regulations are barriers for selecting land transport. However, cross-border land corridors contribute to the increase of land ratio.

**Keywords:** land transport; cross-border land corridor; Tobit model

## 1. Introduction

A land corridor is a type of highway or railway infrastructure that links two or more urban areas [1]. They typically comprise one or more routes that connect economic centers within and across countries [2]. Several cross-border corridor projects have been implemented by international agencies with the help of worldwide donors to facilitate trade via land transport. These projects have been initiated not only in developed areas, such as the European Union (EU) and North America, but they have also taken place in relatively underdeveloped areas, such as Africa and Asia [2]. The Chinese government is forging ahead on their Belt and Road Initiative, which will connect Europe to China via cross-border land ("belt") and maritime ("road") corridors [3]. Economic corridors in the Greater Mekong Subregion (GMS) (e.g., North–South, East–West, and Southern Economic Corridors designated as development priorities), as initiated by the Asian Development Bank, were formed to accelerate subregional development [4], and to review their configuration to enhance effectiveness and efficiency for advancing its economic integration [5]. These projects are expected to primarily benefit shippers, freight forwarders, and carriers by reducing transport time and costs along the corridors via infrastructure developments and improvements by institutions which are supported by regulations (e.g., cross-border transport agreements and customs clearances) for seamless transport.

Maritime transport is sometimes considered to be superior to land transport for long-distance inter-regional cross-border trade, because the costs are less, owing to economies of scale that come from loading a large amount of cargo, although it is much slower than

land transport. Maritime transport also has an advantage in terms of energy efficiency and greenhouse gas emissions as it is a more environmentally sustainable transport mode than land transport [6,7]. In contrast, land transport has a disadvantage in that crossing borders may spend more time and cost. Furthermore, the capacity per shipment is smaller in land transport than that in maritime transport. Moreover, in developing countries, the value of many trade commodities is not as high as that found in developed countries. Thus, those shippers and forwarders tend to opt for maritime transport instead of land-based transport.

However, in long-distance inter-regional cross-border trade, land and maritime transport may have a competitive relationship, such as the one that led to the Belt and Road Initiative. Land transport has the potential to positively impact economic activities in adjacent areas. Nonetheless, the factors behind choosing land transport have not been clarified at the level of inter-regional cross-border trade even though several studies have done at the intra-regional level. An investigation into the factors that facilitate the selection of cross-border transport via land transport would be useful for donors and agencies, because they need to consider modern and realistic costs vs. benefits regarding transport infrastructure investment (e.g., highway and railway).

This study aims to identify the significant factors that lead to the selection of land corridor transport instead of maritime transport in long distance inter-regional cross-border transport, but not in intra-regional transport. We examine the possible factors by developing a model that estimates land ratio, which is defined as the ratio between the nominal value of land transport and the sum of the land and maritime transport based on data of obtained from the World Trade Service (WTS). We apply a Tobit model with country origin–destination (OD) pairs to sample data. Furthermore, we accumulate as many country OD pairs as possible to improve the results. Land ratios express modal share and are key indicators that explain how modes compete, which can then be used for demand forecasting to predict international cargo values. We also discuss the effects of cross-border land corridors and the possible barriers to selecting them by examining data summaries of the selected OD pairs and residual analyses of Tobit results.

The remainder of this study is organized as follows. Section 2 provides a literature review that deals with the cross-border land corridor and maritime transport by evaluating their performances based on key factors. Section 3 describes the data sources and how we handle data samples. Section 4 describes in detail the Tobit model and 11 explanatory variables that support decision-making. The estimated results are then discussed in Section 5. Section 6 concludes this article.

## 2. Literature Review

Several studies have evaluated cross-border land and maritime transport problems by evaluating their performance based on key factors. Moon et al. [8] performed a comparative analysis of selected sea and land transport routes between the Republic of Korea and Europe using the TOPSIS technique. This technique uses quantitative factors that include transport distance, time, and cost, and qualitative factors that include reliability, flexibility, frequency, information service, and safety. Transakul et al. [9] used an analytic hierarchy process to evaluate factors that facilitated cross-border trade. They set cost, time, and complication as the intermediary factors with transparency, technology, policy, and infrastructure as sub-factors, and concluded that transparency was of the greatest importance at the sub-factor level. Banomyong [10] developed the logistics macro-level scorecard based on the four components of a logistics system (i.e., infrastructure, institutional framework, service providers, and traders) to evaluate a system's capability in terms of strengths and weaknesses. With this scorecard, the author benchmarked the North–South and the East–West Economic Corridors of the GMS from physical and non-physical aspects. Four stages of land economic corridor development were also proposed. Regmi and Hanaoka [11] used a time–cost–distance method to highlight the importance of improving hard and soft infrastructures to assess two important intermodal transport corridors linking north, east, and central Asia. Conditions of transport infrastructure, facilities, and clearance processes

at ports and border crossings emerged as significant constraints to intermodal transport operations along the corridors. Transport time and costs were sometimes affected by poor infrastructure quality (e.g., only one railway track, lack of locomotives, and low operational frequency of freight trains). Jain and Jehling [12] described a multi-method approach that involved spatial and non-spatial analyses to investigate disparities along a proposed corridor and examined its integration within the existing settlement structure. They found that the policies affecting transport corridors had a risk of leaving peripheral areas marginalized. Fraser and Notteboom [13] applied the resource and capability approach to the context of corridors connecting a port system to contestable hinterlands in southern Africa. Capacity expansions of seaport and corridor networks as resources in conjunction with efficient transport services/operations as capabilities were found to be important to guaranteeing the attractiveness of port–corridor combinations. Rodemann and Templar [14] conducted a study on intercontinental rail transport between Asia and Europe by performing a literature review with interviews. They identified enablers and inhibitors of intercontinental rail freight using the PESTLE analysis. Enablers for rail transport included investment into transport infrastructure, transport capacity, transport reliability, high security, intergovernmental agreement, geography, climate, $CO_2$ emissions, and energy consumption. Lim et al. [15] derived relevant factors to be considered in the development of transit trade corridors and derived eight underlying factors that affected their design: development and policy implications; safety, security, and political concerns; environmental protection; financing and investment; soft infrastructure; hard infrastructure; geography and landscape; and corridor performance. Wiegmans and Janic [16] proposed a methodology for assessing the performance of long-distance intercontinental intermodal rail/road and sea shipping freight transport corridors and assessed their performance between China and Europe as part of the Belt and Road Initiative. These performances were assumed to be dependent of infrastructural and technical/technological capabilities, such as railway lines, intermodal terminals, rolling stock, and support facilities and equipment. Panagakos and Psaraftis [17] proposed a methodology for freight corridor performance monitoring using key performance indicators. Zhang et al. [18] investigated cold chain-mode choices between containerized transport and reefer bulk shipping. Chain-mode is used to schedule shipments by considering different reefer bulk planning methods, sailing speed optimizations, cargo value depreciation, and greenhouse gas emissions. Wang et al. [19] analyzed the effect of green logistics on international trade using an augmented gravity model and the data of 113 countries and regions over the period of 2007–2014. The explanatory variables were classified into seven categories: economic, environmental, trade facilitation, geographical, political, cultural, and entry cost. Göçmen and Erol [20] examined the allocation of export containers to transport modes in Turkey and European countries by incorporating social risks, such as human accidents and deaths, and ecological risks, such as emissions and noise pollution. A mixed-integer programming-based mathematical model with a fuzzy-based approach was proposed to decide the containers allocation. Tadić et al. [21] evaluated of the location selection and development of dry port terminals as a prerequisite for the establishment of an ecological, economic, and socially sustainable logistics network. In their study, the infrastructure criteria, which included distance and transport, indirectly considered the ecological and economic sustainability of the dry port locations. A new hybrid model of multicriteria decision-making was then developed.

A few studies investigated mode choices between cross-border land and maritime transport. Feo et al. [22] and Arencibia et al. [23] applied the discrete choice model to find freight shipper preferences based on attributes of cost, transit time, punctuality, and service frequency, resulting in the development of explanatory variables related to the different transport modes. A stated preference survey was used for data collection, because it required fewer data. Jiang et al. [24] explored current and prospective hinterland patterns of the China Railway Express using the binary logit model and assessed the impacts of fixed utility, freight costs, and transport time. They determined that shippers were more inclined to select maritime shipping because of the high value of fixed utility. Li et al. [25]

analyzed the influence of geographical factors on the land ports-of-entry (POEs) and sea POEs cross-border logistics routes choice in China, Myanmar, and Vietnam by using the conditional logit model. The results show that there are significant differences in the characteristics and their influencing factors, which are reflected in the scale of freight, distance, duration, transport expense, infrastructure quality, geographical location, and characteristics of the shippers and POEs. Baindur and Viegas [26] used an agent-based modeling approach to find the important factors for mode choices by shippers.

Table 1 summarizes a review of the literature and indicates the methodology, regions or countries, transport mode, and with or without border crossing. This study focuses on OD pairs from the countries and areas in the world where land and maritime transport can compete in long distance inter-regional but not intra-regional cross-border transport that is different from other studies.

Substantive research on the evaluation of land corridors have been conducted, but they vary by the purposes, methods, and sizes of the corridor examined. They also allow for quantitative and qualitative factors to be considered simultaneously. However, it is not still clear how to deal with performance indicators related to cross-border land and maritime transport while considering the influence of the differences of freight volume. Additionally, all of the related studies dealt with specific corridors, and there were none that generalized the factors for worldwide use. Thus, this study develops a model to identify the factors that lead to the selection between the two given transport modes with built-in inter-regionality and worldwide applicability.

**Table 1.** Summary of literature review.

| Papers | Methodology | Regions or Countries | Transport Mode | Border Crossing |
|---|---|---|---|---|
| Moon et al. (2015) [8] | TOPSIS analysis | Sea and land transport routes between the Republic of Korea and Europe | Inland and maritime | √ |
| Transakul et al. (2013) [9] | Analytic hierarchy process | East–West Economic Corridor of Greater Mekong Subregion | Inland | √ |
| Banomyong (2008) [10] | Logistics Macro-Level Scorecard | North–South and the East–West Economic Corridors of Greater Mekong Subregion | Inland | √ |
| Regmi and Hanaoka (2012) [11] | Time-cost-distance method | Important intermodal transport corridors linking North-East and Central Asia | Inland | √ |
| Jain and Jehling (2020) [12] | Spatial and non-spatial analysis | Delhi–Mumbai Industrial Corridor in India | Inland | |
| Fraser and Notteboom (2014) [13] | Resource and capability corridor appraisal model | Corridors connecting a port system to contestable hinterlands for southern Africa | Inland | |
| Rodemann and Templar (2014) [14] | PESTLE analysis | Intercontinental rail transport between Asia and Europe | Inland | √ |
| Lim et al. (2017) [15] | Exploratory and confirmatory factor analysis | Transit trade corridors in the Northeast Asia region | Inland | √ |
| Wiegmans and Janic (2019) [16] | "what if" scenario approach | Intercontinental freight transport corridors spreading between China and Europe | Inland and maritime | √ |
| Panagakos and Psaraftis (2017) [17] | Key Performance Indicators estimation | Green Corridor in the North Sea Region | Inland and maritime | √ |
| Zhang et al. (2020) [18] | Mixed integer linear programing model | (Numerical example only) | Maritime | |
| Wang et al. (2018) [19] | Augmented gravity model | 113 countries and regions all over the world | Unspecified | √ |
| Göçmen and Erol (2018) [20] | A mixed-integer programming-based mathematical model with a fuzzy-based approach | Between Turkey and Europe | Inland and maritime | √ |
| Tadić et al. (2020) [21] | Hybrid multicriteria decision-making model combined Delphi, AHP, and CODAS methods in a grey environment | Western Balkans region | (Dry port) | |
| Feo et al. (2011) [22] | Discrete choice model | Door-to-door road transport and short sea shipping in the Sea of south-west Europe | Inland and maritime | √ |
| Arencibia et al. (2015) [23] | Discrete choice model | Freight flows between Spain and Europe. | Inland and maritime | √ |
| Jiang et al. (2018) [24] | Discrete choice model | China Railway express focusing on its hinterland patterns | Inland and maritime | |
| Li et al. (2020) [25] | Conditional logit model | China, Myanmar, and Vietnam | Inland and maritime | √ |
| Baindur and Viegas (2011) [26] | Agent-based modeling | Atlantic–Mediterranean Transition Region | Inland and maritime | √ |

### 3. Data

A database that provides the nominal trade value of transport modes in OD pairs by country is needed to identify the important factors that facilitate the selection of land transport. The data sorted by commodity and mode can be partially obtained by examining those provided by the statistical institutions of some countries. However, many countries do not have an adequate amount of data. Therefore, we use the WTS data provided by IHS Markit, Ltd. [27].

The WTS provides data from 75 countries and 31 areas. The areas consist of two or more countries classified by 11 regions. For example, one area in western Africa includes Burkina Faso, Mali, and Niger, and another area in Central America includes El Salvador, Honduras, and Nicaragua. All data items, except total trade nominal value, exclude intra-European trade. Therefore, intra-European trade cannot be selected as a sample in this study.

We extract the OD pairs from the countries and areas in which land and maritime transport can compete. First, the countries and areas are selected based on the following four criteria: (1) areas consisting of four or more countries that are excluded, because it is difficult to express the representative values among four or more countries to calculate the explanatory variables; (2) areas containing both landlocked and non-landlocked countries that are excluded, because landlocked countries have no sea port in their territory and the selection conditions are too different; (3) the countries in an area that are physically connected by land; and (4) countries or areas that do not connect with a land border to other countries or areas that are excluded, because they have no opportunity to select land transport (e.g., an island country).

Next, the OD pairs are selected using the following additional two criteria: (5) origins and destinations that are located in the same or a nearby region and (6) land ratio or maritime ratios that are greater than 1% in an OD pair. The term "nearby region" used in criterion (5) indicates regions that geographically neighbor each other. For example, freight transport between China and Germany is not examined in this study, because the two countries are not adjacent, and the distance between them is extremely large. In this case, maritime transport would be superior to land transport. Regarding criterion (6), maritime ratio includes the nominal maritime values over the sum of those of land and maritime. If the land or maritime ratio is less than 1%, the two transport modes are no longer considered to be competing.

The pairs of regional classifications examined as inter-regional cases in this study are listed in Table 2. Pair 5 is an intra-regional pair between EU and non-EU countries, but we decided to include it because the border control exists between these countries. Based on the results of the described selection process, we used 280 OD pairs comprising 64 countries and nine areas (i.e., Benin and Togo; Burkina Faso, Mali, and Niger; Belize and Guatemala; Costa Rica and Panama; El Salvador, Honduras, and Nicaragua; Latvia, Estonia and Lithuania; French Guiana, Guyana and Suriname; Iran and Iraq; and Oman and Yemen) for the Tobit-model analysis.

**Table 2.** Selected regional pairs for the inter-regional cases.

| | | |
|---|---|---|
| Pair 1 | Africa | Western Asia |
| Pair 2 | Central America and the Caribbean | North America |
| Pair 3 | Central America and the Caribbean | South America |
| Pair 4 | East Asia | Indian Subcontinent |
| Pair 5 | European Union | Other European Countries |
| Pair 6 | European Union | Western Asia |
| Pair 7 | Indian Subcontinent | Western Asia |
| Pair 8 | Other European Countries | Western Asia |

## 4. Methodology

We used a Tobit model to find the important factors that facilitate the selection of land transport. We applied land ratio, which has upper and lower limits, as a dependent variable. This way, the Tobit model should yield more desirable results than the ordinary least-squares method, which is otherwise widely applied when the range of the dependent variables is limited.

The Tobit model is defined by Equation (1). The model aims to determine the best-fitting model to describe the relationship between the dependent variable, $y$, and the set of explanatory variables, $x_{1.i}, \dots, x_{p.i}$, by estimating the intercept, $\alpha$, the coefficients, $\beta_1, \dots, \beta_p$, and their significance levels. The Tobit model supposes that there is a latent variable, $y_i^*$, expressed as a linear combination of explanatory variables, intercepts, and coefficients. Additionally, there is a normally distributed error term, $u_i$, which is used to capture random influences on the relationship between explanatory and dependent variables. The dependent variable is equal to the latent variable whenever the latent variable is in the range of zero to one:

$$y_i = \begin{cases} 0 & if & y_i^* \leq 0 \\ y_i^* & if & 0 < y_i^* < 1 \\ 1 & if & y_i^* \geq 1 \end{cases}, \tag{1}$$

where $y_i^*$ is a latent variable: $y_i^* = \alpha + \sum_{j=1}^{p} \beta_j x_{j.i} + u_i.$

$$Land\ Ratio[\%] = \frac{Overland\_Other\ Trade\ Nominal\ Value\ (USD\ Thousands)}{Total\ Trade\ Nominal\ Value\ (USD\ Thousands) - Airborne\ Trade\ Nominal\ Value\ (USD\ Thousands)}. \tag{2}$$

The total trade nominal value of the WTS data has three components: overland/other trade nominal value, seaborne trade nominal value, and airborne trade nominal value. Overland/other trade nominal value indicates land transport. We use Equation (2) to express the land ratio to best show the ratio of land transport to the total trade nominal value. Air transport offers high-speed deliveries at high costs. Thus, airborne trade is generally limited to specific commodities having high value and small size, such as electronic components, semiconductors, computer equipment, and parts. However, the difference between the defined land ratio and the land ratio without excluding airborne trade value from the denominator is less than 1% in 224 of the 280 samples, because we excluded the OD pairs covering large distances by criterion (5) of the OD pair selection process. In relatively short-distance cases, road transport has a similar function as air transport for offering high-speed transport at high costs. Therefore, excluding the airborne trade value does not significantly affect the results.

In this study, we selected 11 explanatory variables based on several existing studies. For each variable, we next explain the definition, selection justification, calculation method, and the year to be used for its calculation. Note that, if either the origin or destination is an area, the mean is used for the input.

- Gross Domestic Product (GDP) per capita. The definition of GDP per capita (USD) is the GDP divided by mid-year population. It is used in our model, because it expresses a proxy value of the economic power of shippers or forwarders who can pay the transport costs. Moreover, our target countries have different economic powers that are related to the transport cost as a typical factor for mode choice [22,23]. The cost of land transport is generally higher than that of maritime transport, and a country wielding economic power can almost always afford to use the mode having the higher cost. Thus, the expected sign is positive. We calculate the mean of the origin and destination country/area as the input value of the OD pair, because the shipper or forwarder who pays the transport cost between the OD countries/areas depends on the trade contract [28].

- Distance. The distance is measured by connecting the main cities in the origin country/area and the destination country/area with the main cities of the country/area in-between the two. The cost per distance of road and rail transport was higher than that of maritime transport, whereas the road and rail transport have a lower cost for short distances [29]. The distance variable represents the degree to which maritime transport is superior to land transport from a cost perspective. Thus, the expected sign is negative. Distance data were taken from the maps prepared by the Geospatial Information Authority of Japan [30] and are presented in kilometers. We selected the capital city as the main city. However, the largest economic city was selected in some countries. If the origin or destination was an area, the city used to measure the distance was selected as the main city of the highest GDP country of the included countries.

- Export of manufacturing commodity. The fact that transport flows were highly heterogeneous is undoubtedly a critical aspect when analyzing freight transport [31]. The value of the cargo being transported was included as heterogeneous and can affect the choice of transport mode. We used the ratio (percentage) of manufacturer exports over the total amount of merchandise exports from the origin country/area. We assumed that manufacturing products had a higher value than did other products, such as agricultural raw materials and fuels. Thus, the expected sign was positive. We selected Sections 5 (chemicals), 6 (basic manufactures), 7 (machinery and transport equipment), and 8 (miscellaneous manufactured goods) and excluded division 68 (non-ferrous metals) as manufacturing commodities from the Standard International Trade Classification [28].

- Landlocked country (dummy). A landlocked country does not have a port in its territory. Thus, it often requires more cost and time to trade [32,33]. The dummy variable is equal to one if the origin or destination country is a landlocked country. Otherwise, it is set to zero. The expected sign is positive, because land transport should be superior to maritime transport.

- Neighboring country/area (dummy). This dummy variable is set to one if the origin and destination country/area shares a land border. Otherwise, the variable is set to zero. We assume that a border crossing is an obstacle to land transport, because the cargo must pass through customs, immigration, and quarantine, which leads to longer transport times and higher costs [11]. However, if an OD pair neighbors the country, the number of border crossings can be only one. Thus, the expected sign is positive.

- Number of land borders. This variable represents how many times the cargo must cross a border when being transported by road or rail. This represents an additional obstacle during land transport. Thus, the expected sign is negative.

- Country risk. Euromoney [34] calculates country risk by conducting a consensus survey of expert opinions from 186 countries. The scores express a social network of economic and political risk for each country. We assume that the country risk affects land transport, especially at borders where ethical conflicts, corruption, or bribes may occur [35]. The scores range from 0 to 100. A higher score indicates a lower country risk value. Thus, the expected sign is positive. We calculate the mean of the origin and destination country/area as the input value of the OD pair. The year of data used in this study is 2011.

- Infrastructure level. The level of investment in land infrastructure (e.g., highways and railways) affects the time and cost of land transport. Most research in the literature review [9,10,13] assumed it to be a critical factor. Thus, we use three indicators to express the road conditions between the origin and destination. First is the ratio of total road length to total land area ($km/km^2$). Second is the ratio of paved road length to total road length (percentage). Third is the ratio of total railway length to total land area ($km/km^2$). These data were derived from The CIA World Factbook [36]. Many studies integrated indicators that used principle component analysis, wherein the first principle component is used as the input value [33,37]. We also implemented three indicators, and the first principle component held more than 60% of the variable

information. Therefore, we use the mean of the first principle component of the origin country/area and destination country/area as the input value of the OD pair, which has both positive and negative values. More land infrastructure thus provides better conditions for land transport. Thus, the expected sign is positive.

- Port access time. Transport time is a significant factor for mode choice [22,23]. Thus, port access time, a component of transport time, is used to represent the port accessibility of a major city. We use the sum of the export lead time in an origin country/area and the import lead time in the corresponding destination country/area as the input value of the OD pair. The unit is a day [38]. The definition of export lead time is the median time (the value for 50% of shipments) from shipment point to port of loading, and the import lead time is the median time (the value for 50% of shipments) from port of discharge to arrival at the consignee. Longer port access time is, therefore, better for land transport. Thus, the expected sign is positive.

- Port infrastructure level. Port infrastructure level represents the development level of port infrastructure in the origin and destination countries/areas. This is a critical factor for the same reason as the infrastructure level. These data reflect the quality of port infrastructure [38]. This index ranges from 1 to 7 and measures the perception of a country's port facilities as assessed by business executives. Because higher values indicate better port quality, the expected sign is negative. We calculate the mean of the origin and destination countries/areas as the input value of the OD pair. In the case of a landlocked country, we use the index value of the quality of port of the country that the landlocked country generally used at the import and export [39].

- Maritime transport frequency. It was compared the performances of rail/road and sea shipping freight transport corridors in terms of the transport service frequency in operational performances [16]. Thus, we use the linear shipping connectivity index as the maritime transport frequency between the origin and destination countries/areas [38]. This index captures how well countries are connected to global shipping networks based on five components of the maritime transport sector: number of ships, container-carrying capacity, maximum vessel size, number of services, and number of companies deploying container ships in a port. The index generates a value of 100 for the average index. Therefore, the values range from zero to more than 100. Because higher values better reflect maritime transport services, the expected sign is negative. We calculate the mean of the origin and destination as the input value of the OD pair. For a landlocked country, we use the general index value of the country of the landlocked country, similar to the port infrastructure-level variable.

We selected the pass-through countries/areas to calculate the values of distance, number of land borders, infrastructure level, and country risk when the origin and destination countries/areas did not neighbor each other and had two or more connected countries/areas between them. For example, if the origin country was Thailand, and the destination country was Vietnam, Cambodia was the pass-through country. Likewise, if the origin country was Nigeria, and the destination country was the Ivory Coast, the pass-through countries were Benin, Togo, and Ghana. We determined the pass-through countries by the following priorities. First was the route having a land corridor developed by a regional or international donor based on the study of Arnold [2], starting from the corresponding country or the neighboring country. Second was the route having the fewest border-point crossings, and third was the route having the shortest distance.

The overall summary of variables is given in Table 3. As seen in Table 4, there is no pair having a correlation coefficient greater than 0.7 between explanatory variables. Therefore, all variables were used to conduct the Tobit model analysis in the next section.

**Table 3.** Summary of variables.

| Variables | Max | Min | Mean | Standard Deviation | Expected Sign |
|---|---|---|---|---|---|
| Land ratio [%] | 98.96 | 1.07 | 42.80 | 37.39 | |
| GDP per capita [USD] | 70,728 | 853 | 15,918 | 14,776 | + |
| Distance [km] | 7728 | 81 | 2496 | 1651.4 | − |
| Export of manufacturing commodity [%] | 94.0 | 0.1 | 44.8 | 30.9 | + |
| Landlocked country | 1 | 0 | 0.11 | 0.32 | + |
| Neighboring country/area | 1 | 0 | 0.33 | 0.47 | + |
| Number of land border | 12 | 1 | 2.98 | 2.25 | − |
| Country risk [index] | 84.64 | 29.80 | 53.23 | 11.18 | + |
| Infrastructure level [index] | 2.0333 | −1.3722 | −0.1895 | 0.8909 | + |
| Port access time [days] | 16.0 | 2.0 | 5.4 | 2.6 | + |
| Port infrastructure level [index] | 6.7 | 2.7 | 4.3 | 0.7 | − |
| Maritime transport frequency [index] | 139.1 | 4.7 | 40.7 | 23.2 | − |

**Table 4.** Correlation coefficients between the explanatory variables.

| | 1 | 2 | 3 | 4 | 5 | 6 | 7 | 8 | 9 | 10 | 11 |
|---|---|---|---|---|---|---|---|---|---|---|---|
| 1 GDP per capita | 1.00 | | | | | | | | | | |
| 2 Distance | 0.15 | 1.00 | | | | | | | | | |
| 3 Export of manufacturing commodity | 0.01 | 0.21 | 1.00 | | | | | | | | |
| 4 Landlocked country | −0.09 | −0.10 | −0.01 | 1.00 | | | | | | | |
| 5 Neighboring country/area | −0.18 | −0.38 | −0.26 | 0.01 | 1.00 | | | | | | |
| 6 Number of land border | 0.41 | 0.56 | 0.33 | −0.04 | −0.61 | 1.00 | | | | | |
| 7 Country risk | 0.67 | 0.09 | 0.04 | −0.09 | −0.04 | 0.14 | 1.00 | | | | |
| 8 Infrastructure level | 0.43 | 0.17 | 0.61 | 0.09 | −0.33 | 0.57 | 0.37 | 1.00 | | | |
| 9 Port access time | −0.03 | −0.22 | −0.53 | 0.06 | 0.32 | −0.31 | −0.49 | −0.19 | 1.00 | | |
| 10 Port infrastructure level | 0.66 | 0.21 | 0.29 | −0.10 | −0.27 | 0.49 | 0.58 | 0.50 | −0.37 | 1.00 | |
| 11 Maritime transport frequency | 0.18 | 0.28 | 0.56 | −0.01 | −0.10 | 0.16 | 0.46 | 0.33 | −0.39 | 0.52 | 1.00 |

## 5. Results and Discussion

### 5.1. Summary of the Land Ratio

If an OD pair locates neighbors connected by a land border, it may be advantageous to select land transport over maritime, as explained by the neighboring dummy. Figure 1 presents the average value of the land ratio dividing neighbors and non-neighbors in each region. The average land ratio of neighbors is higher than non-neighbors among the OD samples in all regions. Therefore, we understand that border crossing is a bottleneck for land transport, because the cargo must cross the borders more than once if the OD pairs are non-neighbors. For neighbors, the average values of the land ratio in the region are greater than 50%, except for east Asia and South America.

In east Asia, the only five samples of neighboring OD pairs are Malaysia–Thailand (57.24%), and vice versa (57.80%); China–Vietnam (32.52%), and vice versa (33.50%); and Singapore–Malaysia (53.56%). The land ratio of China–Vietnam is lower than most neighboring OD pairs in other regions, because the main industrial cities in China (e.g., Shanghai) are located in northern coastal areas, and the main Vietnamese industrial city, Ho Chi Minh City, is located in southern Vietnam. Thus, land transport has few advantages. Malaysia–Singapore was excluded, because this OD pair does not satisfy criterion (6) of the OD pair selection, which is less than 1% of the maritime ratio. This is understandable, because Singapore is an island country, and the two are connected by the Johor–Singapore Causeway. The reason why the land ratio of Singapore–Malaysia is only 53.56% is that

Malaysian vehicles can drive in the territory of Singapore without permission. However, only permitted Singapore vehicles, including trucks, can enter the territory of Malaysia based on Malaysian law [40]. Therefore, maritime transport is also used from Singapore to Malaysia.

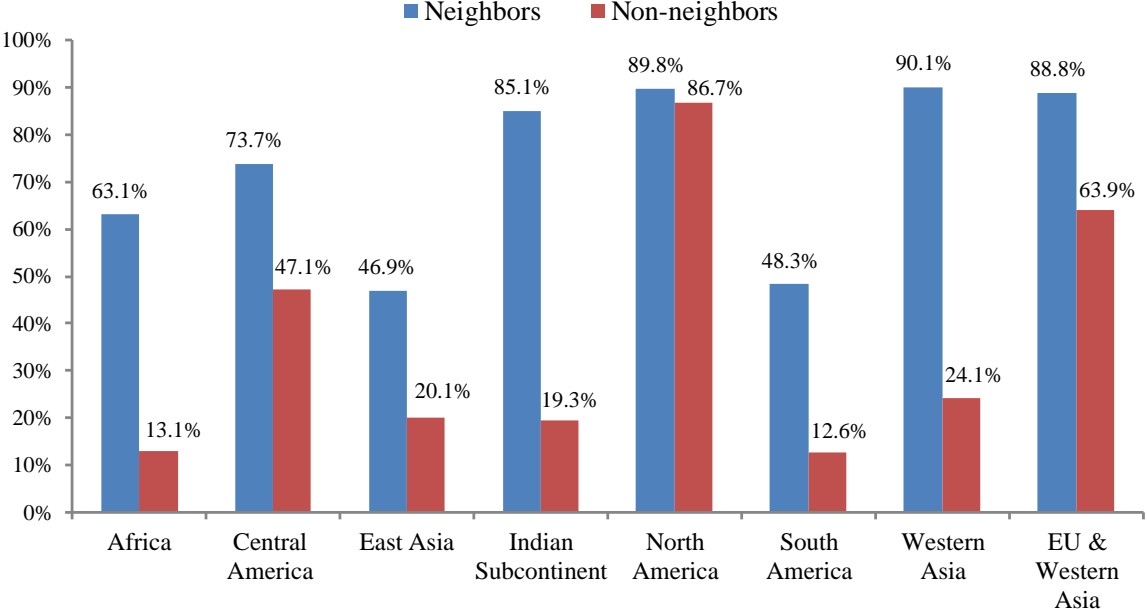

**Figure 1.** Average land ratio of neighbors and non-neighbors by region.

In South America, there are 26 samples of neighboring OD pairs. Among them, the land ratios of 10 OD pairs were between 12 and 24%, including Chile–Peru, Argentina–Chile, Brazil–Peru, Brazil–Venezuela, Brazil–French Guiana/Guyana/Suriname, and the reversals of each. Therefore, the average value is less than 50%. Geographical conditions of borders in these OD pairs are very tough to pass by truck, because the Andes Mountains and the Amazon rainforest exist along the borders. In Africa, the land ratio of some neighboring OD pairs is extremely low, because the Sahara Desert exists between them. As described later, such geographical conditions affect the gap between the actual land ratio and that predicted by the estimated model.

Among the non-neighboring OD pairs, the average values of the land ratios in North America, EU, and western Asia (i.e., Middle East countries except Egypt and including Turkey, following the WTS definition) were greater than 50%, because most countries of these areas are developed. Thus, shippers and forwarders may select land transport with relatively higher tariffs. Additionally, these countries have better quality land infrastructure. Among the EU countries, there are no physical and institutional barriers, which can encourage the use of land transport.

The reason for the high land ratio in the EU and western Asia is that Turkey is either the origin or the destination country in western Asia. It has a high land ratio with many non-neighboring European countries, such as the Czech Republic (95.39 and 95.09), Hungary (95.27 and 94.19), and even Romania (95.16 and 95.26) connected through the Black Sea. The land ratio between Turkey and The Netherlands, Belgium, Portugal, and four north European countries have relatively lower values: ~79–80%.

Some cross-border land corridors may contribute the higher land ratio among non-neighboring OD pairs. In North America, the CANAMEX corridor connects Canada and Mexico through the United States by a series of highways under the North American Free Trade Agreement. Thus, both Canada–Mexico (85.87%), and vice versa (87.63%), have high land ratios. In Central America, although this region is divided into three areas without a single country as an independent OD, there are mid-land ratio values

between Belize/Guatemala–Costa Rica/Panama (47.02%), and vice versa (47.09%). However, Mexico–El Salvador, Honduras, and Nicaragua (89.03%), and vice versa (89.29%), and Mexico–Costa Rica and Panama (78.92%), and vice versa (83.46%), have high land ratios, owing to their connections by the Pan-American Highway. In east Asia, Thailand and Vietnam are not neighboring countries, with Cambodia between them. However, the land ratio is relatively high, as with Thailand–Vietnam (66.56%), and vice versa (65.02%), because the Southern Economic Corridor, one of the GMS corridors, may contribute to the land connectivity between Ho Chi Minh City and Bangkok.

### 5.2. Model Estimation

The results of the Tobit model estimation performed for all 280 samples are summarized as the base case in Table 5. We obtained that distance, export of manufacturing commodity, landlocked country, neighboring country/area, number of land borders, country risk, infrastructure level, port access time, and maritime transport frequency were significant. Additionally, the expected sign, apart from a few land borders, GDP per capita, and port infrastructure level, was not significant.

**Table 5.** Results of Tobit model estimation.

| Variable | Base | | Exclude Two European Inter-Regions | | Neighbors (without Landlocked Countries) | | Non-Neighbors | |
|---|---|---|---|---|---|---|---|---|
| | Coeff. | Std Errors | Coeff. | Std Errors | Coeff. | Std Errors | Coeff. | Std. Errors |
| Intercept | −0.2698 | 0.1941 | −0.4002 * | 0.2117 | 0.0533 | 0.3135 | −0.0613 | 0.2358 |
| GDP per capita | −0.0000 | 0.0000 | 0.0000 | 0.0000 | −0.0000 | 0.0000 | −0.0000 * | 0.0000 |
| Distance | −0.0001 *** | 0.0000 | −0.0001 *** | 0.0000 | −0.0001 *** | 0.0000 | −0.0001 *** | 0.0000 |
| Export of manufacturing commodity | 0.0030 *** | 0.0008 | 0.0027 *** | 0.0008 | 0.0024 * | 0.0014 | 0.0032 *** | 0.0008 |
| Landlocked country | 0.2169 *** | 0.0476 | 0.0735 | 0.0674 | - | | 0.2112 *** | 0.0542 |
| Neighboring country/area | 0.2799 *** | 0.0400 | 0.3155 *** | 0.0472 | - | | - | |
| Number of land border | 0.0462 *** | 0.0123 | −0.0196 | 0.0194 | - | | 0.0358 *** | 0.0123 |
| Country risk | 0.0106 *** | 0.0020 | 0.0018 | 0.0023 | 0.0022 | 0.0039 | 0.0129 *** | 0.0024 |
| Infrastructure level | 0.0559 * | 0.0294 | −0.1381 *** | 0.0378 | −0.0635 | 0.0599 | 0.0940 *** | 0.0339 |
| Port access time | 0.0171 ** | 0.0075 | 0.0046 | 0.0079 | 0.0241 * | 0.0127 | 0.0014 | 0.0093 |
| Port infrastructure level | 0.0291 | 0.0351 | 0.1410 *** | 0.0388 | 0.1057 * | 0.0618 | −0.0307 | 0.0421 |
| Maritime transport frequency | −0.0019 * | 0.0010 | −0.0024 ** | 0.0010 | −0.0013 | 0.0021 | −0.0019 * | 0.0010 |
| Adjusted R squared | 0.5622 | | 0.5396 | | 0.4153 | | 0.6192 | |
| Number of samples | 280 | | 210 | | 81 | | 188 | |

(***; **; * show significance at the 1%, 5%, and 10% level, respectively).

As shown in Figure 1, there are high land ratios between the EU and western Asia, although they have many border crossings. This might affect the estimated results of the base case that includes a significantly positive sign for the number of land borders. Therefore, we examined the Tobit model excluding two European-related cases. One is EU countries and western Asia, and the other is the other European countries and western Asia. The result in Table 4 demonstrates that the number of land borders is not a significant variable, but the sign became negative.

Interestingly, landlocked country, country risk, and port access time are not significant. There are many landlocked countries in Europe, such as Austria, Czech Republic, Hungary, Slovakia, and Switzerland, and these countries have good connections with land infrastructure to both neighbors and non-neighbors. Therefore, the landlocked country variable

became insignificant after removing these European countries from the samples. We can also understand why the number of land borders is significant with the positive coefficient in the base case and becomes insignificant for the same reason. Port access time might also have similar reasons, in that it becomes insignificant, because these European landlocked countries have shorter port access times to Antwerp, Hamburg, and Rotterdam ports. The country risk scores of most European countries are relatively higher than others, indicating lower risk. Thus, country risk may not explain the land ratio after removing the European countries. From this result, we can assume the country risk may not affect the mode choice of developing and emerging countries. The port infrastructure level was significant, but it was opposite of the expected sign.

Next, we examined the Tobit model separated into two groups—OD neighbors and OD non-neighbors—for determining which factors were significant, except for the neighboring effect of each. In the case of OD neighbors, the OD pairs for which the origin or destination was a landlocked country were excluded from the samples, because most have a high land ratio, owing to their borders being connected by land. The number of land borders was also excluded, because the value is one in all samples. The sample size is 81 in the neighboring case and 188 in the non-neighboring case.

The estimated result of the non-neighbors fits more than the neighbors, as indicated by the McFadden's R-squared value; it has a similar result to the base case. In the case of neighbors, only distance was significant at the 1% level. However, export of manufacturing commodity, port access time, and port infrastructure level were significant at the 10% level. This result suggests that distance was a core factor and that infrastructure quality was important to selecting the transport mode between two countries/areas if they are neighbors, but not for landlocked countries.

We discuss the high residual error between the actual and predicted values of the base model for specifying why these occur. Samples having greater than $2\theta$ or less than $-2\theta$ standardized residual error were classified as having high residual errors in this study. All samples having high residual error among OD pairs are summarized in Table 6, where $1\theta$ was 22.4% in the base case in Table 5. It is noted that the predicted values are sometimes greater than 100% or smaller than zero, because these values are the estimated values of the latent variables in Table 6. The reasons why these samples have high residual error can be classified into five factors: geographical conditions, country relationships, regulations, distances, and infrastructure levels. For the first three factors, the predicted values were higher than the actual ones. Thus, we need to find the reasons beyond the 11 explanatory variables.

First, as explained, the actual land ratios in the four OD pairs of North Africa listed in Table 6 are very low, owing to the Sahara Desert. However, each OD pair is a neighboring country/area. There are also the Arabian Desert between Egypt and Bahrain, and the Libyan Desert between Israel and Libya, which might be a barrier to using land transport. We excluded the samples of the two opposite OD pairs, because their land ratios were less than 1%, which does not satisfy criterion (6) of the OD pair selection. In South America, the Darien gap exists in the border area between Panama and Colombia covered by tropical rain forest and river deltas. Therefore, the actual value of the land ratio is low, because the area has a missing link in the Pan-American highways between Central and South America.

Second, political antagonism between two countries might be an obstacle to crossing the land border, because the cargo and drivers are rigorously checked at all stops. Thus, it takes much more time to pass through. The political relations between Egypt and Israel, as well as Pakistan and Iran, are poor. The relation between Israel and Saudi Arabia is also historically poor, and Israel is located between Egypt and Saudi Arabia. Thus, the land ratio is low between them. We excluded the samples of the opposite OD pairs, because their land ratios were less than 1%. Therefore, almost no cargo is transported by land between these countries.

**Table 6.** Samples with high residual error.

| Factors | Region | Origin | Destination | Actual Value | Predicted Value | Residual Error | Std. Residual Error |
|---|---|---|---|---|---|---|---|
| Geographical Conditions | Africa | Algeria | Burkina Faso, Mali, Niger | 4.8% | 55.7% | −50.9% | −2.27 |
| | | Burkina Faso, Mali, Niger | Algeria | 4.1% | 58.0% | −53.9% | −2.40 |
| | | Libya | Burkina Faso, Mali, Niger | 4.1% | 55.9% | −51.8% | −2.31 |
| | | Burkina Faso, Mali, Niger | Libya | 3.8% | 58.8% | −55.1% | −2.45 |
| | Africa and Western Asia | Egypt | Bahrain | 2.9% | 48.5% | −45.5% | −2.03 |
| | | Israel | Libya | 5.1% | 66.2% | −61.1% | −2.72 |
| | Central America and North America | Costa Rica, Panama | Colombia | 7.9% | 61.5% | −53.6% | −2.39 |
| Country Relationship | Africa & Western Asia | Egypt | Saudi Arabia | 6.0% | 54.7% | −48.7% | −2.17 |
| | | Egypt | Israel | 7.1% | 96.0% | −88.9% | −3.96 |
| | | Israel | Egypt | 5.5% | 111.8% | −106.3% | −4.74 |
| | Indian Subcontinent and Western Asia | Pakistan | Iran, Iraq | 3.6% | 68.6% | −65.0% | −2.90 |
| Regulation | East Asia | Singapore | Malaysia | 53.6% | 109.4% | −55.8% | −2.49 |
| Distance | North America | Canada | Mexico | 85.9% | 27.1% | 58.8% | 2.62 |
| | | Mexico | Canada | 87.6% | 39.8% | 47.9% | 2.13 |
| Infrastructure Level | Central America and North America | Costa Rica, Panama | Mexico | 83.5% | 19.4% | 64.1% | 2.86 |
| | | El Salvador, Honduras, Nicaragua | Mexico | 89.3% | 41.3% | 48.0% | 2.14 |
| | South America | Bolivia | Brazil | 98.8% | 48.2% | 50.6% | 2.25 |
| | | Bolivia | Argentina | 92.9% | 45.9% | 47.0% | 2.10 |

Regulation problems between Singapore and Malaysia have been already explained; only permitted vehicles of Singapore can enter the territory of Malaysia.

Finally, the last six OD pairs in Table 6 have high residual error, because their predicted values are much lower than the actual ones. The reason for Canada–Mexico, and vice versa, is the longer distance between them. However, the CANAMEX corridor may contribute to an actual higher land ratio. Regarding last four pairs, the values of infrastructure levels for these OD pairs are lower than the mean values. However, the Pan-American Highway may contribute the high land ratio of the mode choice between Mexico and Central American countries. Additionally, it is inconvenient to select maritime transport for Bolivia, owing to it being a landlocked country.

## 6. Conclusions

We performed a Tobit model analysis to identify the significant factors needed to select land transport for cross-border freight among countries, areas, regions, and cross-regions. Eight variables were identified as significant in the base model: distance, export of manufacturing commodity, landlocked country, neighboring country/area, country risk, infrastructure level, port access time, and maritime transport frequency. The number of land borders was also found to be significant, but was not its expected sign. The case that excluded European countries, distance, exports of manufacturing commodity, neighboring country/area, and maritime transport frequency was significant. The landlocked country variable became insignificant after removing European countries, because European landlocked countries have good connection via land infrastructure to not only neighbors, but also to non-neighbors. We can also conclude that the country risk calculated by Euromoney may not affect the mode choice in developing and emerging countries. In the case of selecting sample ODs from neighbors, apart from landlocked countries, distance was the most significant factor of selecting land transport. Infrastructure quality was also important, but most other factors were insignificant. On the other hand, the estimated result of the non-neighbors was similar to the base case.

We also statistically analyzed why some OD pairs had higher or lower land ratios than expected and why there were high residual errors between the actual and predicted land ratios. We discovered the reasons, which included geographical conditions; country relationships; regulations for lower actual values than the predicted ones; and the effect of cross-border land corridors in North, Central, and South America and southeast Asia, for higher actual values than the predicted values. These results can contribute to facilitating the development projects of cross-border land corridors for international agencies and donors so that they can carefully consider these factors in the improvement of land transport infrastructure via greater investments in paved roads and railways. This particularly useful to countries and regions that wish to overcome the geographical disadvantages and establish adequate regulations to mitigate vehicle entry permissions. A limitation of this study is that factors related with environmental impact were not considered. Environmental problems can be more significant factors in inter-regional transport if a global carbon tax is implemented in the world without exceptions.

**Author Contributions:** Conceptualization, S.H.; methodology, S.H. and T.M.; validation, S.H.; formal analysis, S.H. and T.M.; investigation, T.M. and W.S.; data curation, S.H. and W.S.; writing—original draft, S.H., W.S. and T.H.; writing—review & editing, S.H., T.M. and T.H.; supervision, S.H. and T.K.; All authors have read and agreed to the published version of the manuscript.

**Funding:** This research received no external funding.

**Institutional Review Board Statement:** Not applicable.

**Informed Consent Statement:** Not applicable.

**Data Availability Statement:** Publicly available datasets were analyzed in this study. This data can be found here: [https://ihsmarkit.com/index.html](https://ihsmarkit.com/index.html); [http://data.worldbank.org/](http://data.worldbank.org/); [http://japonyol.net/editor/distancia.html](http://japonyol.net/editor/distancia.html); [http://www.euromoneycountryrisk.com/](http://www.euromoneycountryrisk.com/); [https://www.cia.gov/the-world-factbook/](https://www.cia.gov/the-world-factbook/); [https://datatopics.worldbank.org/world-development-indicators/](https://datatopics.worldbank.org/world-development-indicators/)].

**Conflicts of Interest:** The authors declare no conflict of interest.

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
