# Peer review of "Identifying Factors for Selecting Land over Maritime in Inter-Regional Cross-Border Transport"

_sustainability, doi:10.3390/su13031471_

Round 1
Reviewer 1 Report
In the era of globalization cross-border transport constitutes a core for an effective distribution of goods. As authors of the manuscript stated “land transport is an alternative choice to maritime transport in cross-border transport” despite being costlier. I find the paper very interesting and worth from the theoretical perspective. Using a Tobit model as an alternative for the ordinary least squares method is an added value for the conducted analysis.
The main aim of the paper is formulated correctly and refers to the identification of “the factors that lead to the selection of these two transport modes”. Authors identified a literature and empirical gap in terms of the factors of inter-regional cross-border transport. Filling that gap would be useful for donors and agencies in terms of the transport infrastructure development. However, in the conclusion section there is a lack of suggestions and ideas on how to use the model prepared by the authors.
I couldn’t agree more with the statement: “most trade statistics offer the nominal value of trade by commodity, but not by transport mode” that is a big challenge for academicians in terms of the quality of their research. The presented method of analysis is characterized in an advanced manner and the exclusion criteria for the areas and pairs selection are widely explained.
I find very interesting the conclusion that “the number of land borders is not a significant variable”. It suggests that Belt and Road Initiative could become a profitable infrastructural investment for cross-border land transport and an alternative for a maritime transport in long distances (e.g. Shanghai – Hamburg, Shanghai – Rotterdam, Shanghai – Antwerp, Shanghai – Gdansk). The sentence “the distance is a core factor and infrastructure quality is important to select the transport mode between two countries/areas if they are neighbours” is a good summary of the manuscript that can be initially pre-assumed.
Nevertheless, there are some issues that should be corrected and/or explained by authors:
– The GMS project seems to be beneficial not only for shippers, freight forwarders and transporters but also for carriers. Maybe the term “transporters” is improper and should be changed to “carriers” (Page 2, line 44).
– Page 2, line 35 – European Union is not a country but a regional association. The same situation is with North America, Africa and Asia in line 36.
– Page 2, line 71 – double “to”: “to to”
– The summary of the literature review in section 2 is too general. I encourage authors of the paper to graphically present the summary informing about the methods and ratios analyzed in the literature review.
– As stated above, in the conclusion section there should be suggestions on how to use the model prepared by authors in order to promote it alongside agencies.
– In the references section, it is necessary to meaningfully incorporate the relevant literature from the journal. There are no references to Sustainability, there so it is not clear on how your research build-upon relevant research in Sustainability.
– According to the instruction for authors, references must be numbered in order of appearance in the text.
– Variables for the model are selected correctly with well-founded justification. However, some of the data are from 2011 or 2014. Why there is no more recent data?
– In the literature review there is no explanation or analogy to one of the existing ratios that assesses the logistics and transport issues in a macro level: Logistics Performance Index (LPI). It is not necessary to include that ratio in the manuscript; however, I would like to know the authors’ opinion on LPI in terms of the cross-border transport.
Concluding, I strongly encourage authors of the manuscript to promote their work around agencies responsible for transport infrastructure development.
Author Response
Dear Reviewer,
We are truly grateful for your comments and thoughtful suggestions. Based on these comments and suggestions, we have made careful modifications on the original manuscript and showed the modified sections in red color on the revised one.
- I couldn’t agree more with the statement: “most trade statistics offer the nominal value of trade by commodity, but not by transport mode”
We have deleted this statement.
- The GMS project seems to be beneficial not only for shippers, freight forwarders and transporters but also for carriers. Maybe the term “transporters” is improper and should be changed to “carriers” (Page 2, line 44).
We have changed to "carriers" in line 38, Page 1.
- Page 2, line 35 – European Union is not a country but a regional association. The same situation is with North America, Africa and Asia in line 36.
We have changed to "areas" in line 29 and 31, Page 1.
- Page 2 line 71 – double “to”: “to to”
We asked an editing company to edit the whole manuscript and updated the expression in English.
- The summary of the literature review in section 2 is too general. I encourage authors of the paper to graphically present the summary informing about the methods and ratios analyzed in the literature review.
We have made and added a new Table 1 as a summary of literature review that indicates methodology, regions or countries, transport mode and with or without border crossing.
- As stated above, in the conclusion section there should be suggestions on how to use the model prepared by authors in order to promote it alongside agencies.
Appreciating this comment, we have added the suggestions in line 554, 555 and 557, Page 17.
- In the references section, it is necessary to meaningfully incorporate the relevant literature from the journal. There are no references to Sustainability, there so it is not clear on how your research build-upon relevant research in Sustainability.
Appreciating this comment and the similar comments of the reviewer 2, we added four papers ([6], [19], [20] and [21]) published in Sustainability.
- According to the instruction for authors, references must be numbered in order of appearance in the text.
We revised the reference to use the number in order of appearance to follow the instruction.
- Variables for the model are selected correctly with well- founded justification. However, some of the data are from 2011 or 2014. Why there is no more recent data?
Originally, the idea of this paper came from the initiation of Belt and Road Initiative (BRI) in 2013, therefore we used the data around the year 2014 before appearing the impact on BRI. However, we have not written these statements in the revised manuscript. If we should do it, we will do it in the next round revision.
Dataset of Euromoney Country Risk was not updated every year in the past. However, we can purchase the annual data of Euromoney Country Risk now, so we understand we should update this data. If the reviewers give us the next round review opportunity, we will recalculate using a new data set.
- In the literature review there is no explanation or analogy to one of the existing ratios that assesses the logistics and transport issues in a macro level: Logistics Performance Index (LPI). It is not necessary to include that ratio in the manuscript; however, I would like to know the authors’ opinion on LPI in terms of the cross-border transport.
Thank you very much for this important comment. LPI is definitely one of the most significant indexes in global logistics and cross-border transport. However, it is difficult to distinguish how the index value affects the mode choice between maritime and land transport in inter-regional cross-border transport. Therefore, we did not use LPI for our study.
Reviewer 2 Report
The study analyses the factors which weigh into selecting road transport over maritime mode in cross-border transport between countries. The topic and methodology is interesting. Yet I have some very serious concerns over the suitability of this paper and formal analysis.
Major comment on analysis:
1) This journal is sustainability. The scope of journal is very relevant to environmental impact of transportation mode choice. Although the authors focus on mode choice in cross border transport, they do not consider emissions (and environmental impact). This a very big drawback for this paper. The paper should consider environmental impact of mode choice otherwise it cannot be considered for publication in this journal. The results should substantially revised.
2) The study only considers superiority of land transport over maritime transport. But, why do you disregard disadvantages of land transport against maritime mode?
3) The study lacks a grounded gap analysis and motivation to work on this problem. There are several papers on similar topics. The contribution of your work is not so clear (line 130-137 can be enhanced).
-- Factors in the sea ports-of-entry and road ports-of-entry cross-border logistics route choice. Journal of Transport Geography, 84, p.102689.
4) Does this study include intra-region transport? If not, this should be clearly noted in the abstract and introduction. Theoretically I tend to think you cannot consider European Union as a single country as 27 member countries have distinct (maritime or road) transport links between them.
Comments on writing in the manuscript;
1) The written English is a bit below top standards. Paper should be carefully checked carefully.
2) Line 35, 36: EU, Africa, Asia are not countries.
3) Line 58-59 is badly written.
4) Line 66: ratio "expresses"
Comments on literature review;
1) There are following important studies about transport (shipping) mode choice in literature. The paper can include and cite following mode choice papers;
-- Hinterland patterns of China Railway (CR) express in China under the Belt and Road Initiative: A preliminary analysis. Transportation Research Part E: Logistics and Transportation Review, 119, pp.189-201.
-- Cold chain shipping mode choice with environmental and financial perspectives. Transportation Research Part D: Transport and Environment, 87, p.102537.
-- Model-based corridor performance analysis–An application to a European case. European Journal of Transport and Infrastructure Research, 17(2).
2) There are many studies focusing on emissions from maritime supply chains. Following studies can be included and cited in the context of road vs maritime emissions:
-- A review of energy efficiency in ports: Operational strategies, technologies and energy management systems. Renewable and Sustainable Energy Reviews, 112, pp.170-182.
-- Decarbonization pathways for international maritime transport: A model-based policy impact assessment. Sustainability, 10(7), p.2243.
Author Response
Dear Reviewer,
We are truly grateful for your comments and thoughtful suggestions. Based on these comments and suggestions, we have made careful modifications on the original manuscript and showed the modified sections in red color on the revised one.
Major comment on analysis:
- This journal is sustainability. The scope of journal is very relevant to environmental impact of transportation mode choice. Although the authors focus on mode choice in cross border transport, they do not consider emissions (and environmental impact). This a very big drawback for this paper. The paper should consider environmental impact of mode choice otherwise it cannot be considered for publication in this journal. The results should substantially revised.
Thank you very much for this important comment. Appreciating the comment, we added the emission matter in Line 45-47 in Page 2. We also added the limitation of this study in Line 557-560 in Page 17, Conclusion to indicate the environmental problems can be more significant factors in inter-regional transport if the global carbon tax is implemented in the world.
Environmental factors such as CO2 emissions are very important as one of performance results of inter-regional cross-border transport. However, our study aims to identify the significant factors that lead to the selection of maritime or land transport, thus we should carefully consider whether the environmental factors influence the choice of two modes.
- The study only considers superiority of land transport over maritime transport. But, why do you disregard disadvantages of land transport against maritime mode?
Appreciating this comment, we have added the disadvantages of land transport in line 47-49, Page 2.
- The study lacks a grounded gap analysis and motivation to work on this problem. There are several papers on similar topics. The contribution of your work is not so clear (line 130-137 can be enhanced).
We highlighted to focus on the long distance inter-regional cross-border in this paper that is different from other studies, described in line 56-57, Page 2 and line 161-164, Page 4. We also added the papers [25] as suggested.
- Does this study include intra-region transport? If not, this should be clearly noted in the abstract and introduction. Theoretically I tend to think you cannot consider European Union as a single country as 27 member countries have distinct (maritime or road) transport links between them.
Our study does not include intra-regional transport, therefore we added the statement in line 62-63, Page 2 and 162-163, Page 4. In the abstract, we expressed "long distance inter-regional cross-border transport" in line 14-15 for highlighting it. However, Pair 5 is an intra-regional pair between EU and non-EU countries, but we decided to include it because the border control exists between these countries, to show in line 210-211, Page 7.
As for intra-EU, WTS data does not include any data of intra-European trade by different transport modes, so that we cannot use it.
Comments on writing in the manuscript;
- The written English is a bit below top standards. Paper should be carefully checked carefully.
- Line 35, 36: EU, Africa, Asia are not countries.
- Line 58-59 is badly written.
- Line 66: ratio "expresses"
Appreciating the comment, we asked an editing company to edit the whole manuscript for improving our English expression. The company also used a format of Sustainability.
Comments on literature review;
- There are following important studies about transport (shipping) mode choice in literature. The paper can include and cite following mode choice papers;
Thank you for suggesting to add the papers in literature. We added [18] and [24] to follow your suggestion ([17] has been included in the original manuscript). In addition, we also added [25] as suggested.
- There are many studies focusing on emissions from maritime supply chains. Following studies can be included and cited in the context of road vs maritime emissions:
Appreciating this comment, we added [6] and [7] in Introduction. We also added [19], [20] and [21] treating the emission from Sustainability.
Round 2
Reviewer 2 Report
The revision has been done flawlessly. The paper is ready to accepted.